

# Searching for phylogenetic patterns of Symbiodiniaceae community structure among Indo-Pacific Merulinidae corals

Sébastien Leveque[1,2], Lutfi Afiq-Rosli[1], Yin Cheong Aden Ip[1], Sudhanshi S. Jain[1] and Danwei Huang[1]

[1] National University of Singapore, Singapore, Singapore
[2] Université de La Rochelle, La Rochelle, Singapore

## ABSTRACT

Over half of all extant stony corals (Cnidaria: Anthozoa: Scleractinia) harbour endosymbiotic dinoflagellates of the family Symbiodiniaceae, forming the foundational species of modern shallow reefs. However, whether these associations are conserved on the coral phylogeny remains unknown. Here we aim to characterise Symbiodiniaceae communities in eight closely-related species in the genera *Merulina*, *Goniastrea* and *Scapophyllia,* and determine if the variation in endosymbiont community structure can be explained by the phylogenetic relatedness among hosts. We perform DNA metabarcoding of the nuclear internal transcribed spacer 2 using Symbiodiniaceae-specific primers on 30 coral colonies to recover three major endosymbiont clades represented by 23 distinct types. In agreement with previous studies on Southeast Asian corals, we find an abundance of *Cladocopium* and *Durusdinium*, but also detect *Symbiodinium* types in three of the eight coral host species. Interestingly, differences in endosymbiont community structure are dominated by host variation at the intraspecific level, rather than interspecific, intergeneric or among-clade levels, indicating a lack of phylogenetic constraint in the coral-endosymbiont association among host species. Furthermore, the limited geographic sampling of four localities spanning the Western and Central Indo-Pacific preliminarily hints at large-scale spatial structuring of Symbiodiniaceae communities. More extensive collections of corals from various regions and environments will help us better understand the specificity of the coral-endosymbiont relationship.

# INTRODUCTION

The modern coral reef is one of the most diverse marine ecosystems on Earth (*Reaka-Kudla, 1997*; *Fisher et al., 2015*), build mainly by the habitat-forming stony corals (Cnidaria: Anthozoa: Scleractinia) with a calcification process driven largely by their intricate symbiosis with photosynthetic dinoflagellates of the family Symbiodiniaceae (*Muscatine & Cernichiari, 1969*; *Stat, Carter & Hoegh-Guldberg, 2006*). The host provides shelter and inorganic nutrients to the endosymbionts (*Muscatine & Porter, 1977*; *Weis et al., 2001*; *Stat,*

Corresponding author
Danwei Huang,
huangdanwei@nus.edu.sg

*Carter & Hoegh-Guldberg, 2006*) in exchange for fixed carbon (*Muscatine & Cernichiari, 1969*; *Weis et al., 2001*; *Yellowlees, Rees & Leggat, 2008*).

Symbiodiniaceae is extremely diverse, with at least nine genus-level clades which contain hundreds of species (*Arif et al., 2014*; *Thornhill et al., 2014*), although fewer than 30 species have been formally named (*LaJeunesse et al., 2018*). The recognition of this remarkable diversity owes in large part to the use of molecular genetic tools that have helped biologists discover and characterise the major clades and subclades (or types) of Symbiodiniaceae (*Rowan & Powers, 1991a*; *Rowan & Powers, 1991b*; *Rowan, 1998*; *LaJeunesse, 2002*). For decades, these clades have been referred to by letters that run from A to I, but genus names have recently been established for most of them, including *Symbiodinium* Gert Hansen & Daugbjerg (2009) (formerly Clade A), *Breviolum* Parkinson & LaJeunesse (2018) (formerly Clade B), *Cladocopium* LaJeunesse & Jeong (2018) (formerly Clade C), and *Durusdinium* LaJeunesse (2018) (formerly Clade D) *LaJeunesse et al. (2018)*. A robust phylogenetic classification of these dinoflagellates is critical for better understanding the coral reef ecosystem because different endosymbiont types have distinct physiological characteristics (*Warner, Fitt & Schmidt, 1996*; *Baker, 2003*; *Sampayo et al., 2008*) that result in broadly predictable responses of reef corals to environmental stress (*Warner, Fitt & Schmidt, 1999*; *Baker et al., 2004*; *Reynolds et al., 2008*). For example, coral hosts dominated by *Durusdinium* can generally better withstand thermal stress compared to corals with a greater abundance of *Cladocopium* (*Rowan, 2004*; *Berkelmans & Van Oppen, 2006*; *LaJeunesse et al., 2014*).

Reef corals exhibit varying levels of specificity for particular genera and types of endosymbionts (*Thomas et al., 2014*; *Smith, Ketchum & Burt, 2017a*). Generally, host species in the Indo-Pacific are mostly associated with the genus *Cladocopium* and to a lesser extent *Durusdinium* (*LaJeunesse et al., 2010a*), though endosymbioses with *Symbiodinium* and *Breviolum* have also been documented (*Loh, Carter & Hoegh-Guldberg, 1998*; *Yang et al., 2012*). Each host colony can also associate with multiple Symbiodiniaceae types (*Rowan & Powers, 1991b*; *Rowan & Knowlton, 1995*; *Baker, 2003*). For example, a single colony of *Porites lutea* can harbour three species of *Cladocopium*—C3, C15 and C91 (*Gong et al., 2018*). Environmental factors may also play a role in the distribution of Symbiodiniaceae among different colonies of a single host species. For instance, the relative abundances of *Cladocopium goreaui* (C1), *Cladocopium* C2, and *Durusdinium glynni* (D1) in *Acropora millepora* are dependent on temperature and light availability (*Cooper et al., 2011*). Dissolved nutrients have also been shown to influence the community structure of the endosymbionts present within the host (*Sawall et al., 2014*; *Gong et al., 2018*).

Apart from the coral host's ability to harbour a wide range of Symbiodiniaceae species, the composition of endosymbionts present in a host population can change in response to environmental variations. For example, following a bleaching event, *Acropora millepora* colonies that were originally dominated by the thermally sensitive *Cladocopium* C3 (=ITS1 type C2; *Tonk et al., 2013*) suffered higher mortality compared with colonies that were *Durusdinium* predominant (*Jones et al., 2008*). Furthermore, the majority of the surviving colonies that were initially *Cladocopium* C3-predominant acquired more *Cladocopium goreaui* or *Durusdinium* endosymbionts (*Jones et al., 2008*). Indeed, the ability to shuffle

and even switch symbiont communities by repopulation of more tolerant types can play an important role towards enhancing the holobiont's recovery capacity and host survivability (*LaJeunesse et al., 2010b*; *Kemp et al., 2014*; *Silverstein, Cunning & Baker, 2015*). Overall, however, the host-symbiont association in the majority of corals tends to be stable, even in the course of severe bleaching events (*Goulet, 2006*; *Smith et al., 2017b*). This stability may play a crucial role in the hosts' resilience during acute environmental stress.

Recently, host identities of populations and species of Pocilloporidae Gray, 1840, have been shown to affect associated Symbiodiniaceae and bacterial communities (*Bongaerts et al., 2010*; *Bongaerts et al., 2011*; *Brener-Raffalli et al., 2018*). Whether this phylogenetic constraint extends beyond intraspecific relationships or sister species is unclear. With rapid advances in our understanding of coral evolutionary history and taxonomy in the past decade (*Frank & Mokady, 2002*; *Budd et al., 2010*), we now can now bear on this question using an abundance of phylogenetic data (*Bhattacharya et al., 2016*; *Kitahara et al., 2016*; *Quek & Huang, 2019*). It is generally acknowledged that gross morphological traits can converge among distinct taxa living under similar environmental influences and these traits evolve at vastly different rates among lineages (*Cairns, 2001*; *Fukami et al., 2004b*; *Flot et al., 2011*), partly explaining why taxonomy based solely on morphology has been problematic (*Fukami, 2008*; *Fukami et al., 2008*; *Kitahara et al., 2010*). However, there exist a number of species traits which are phylogenetically constrained among corals, in that closely-related species tend to possess similar trait states (*Madin et al., 2016a*). These include growth rates (*Madin et al., 2016b*), sexuality (*Baird, Guest & Willis, 2009*), reproductive mode (*Kerr, Baird & Hughes, 2011*), endosymbiont transmission strategy (*Hartmann et al., 2017*), and even susceptibility to and resistance against anthropogenic stressors (*Huang, 2012*; *Huang & Roy, 2013*). The present study extends this line of inquiry to determine if endosymbiont communities are constrained by the phylogenetic relationships among hosts, focusing on a clade of three genera in the family Merulinidae Verrill, 1865, that have a well-characterised phylogeny (*Huang et al., 2014a*). These corals obtain their algal symbionts from the environment (horizontal acquisition), in contrast to the well-studied pocilloporids which vertically transmit their endosymbionts, and thus offer a distinct life history strategy for study (*Baird, Guest & Willis, 2009*; *Hartmann et al., 2017*).

Merulinidae is a clade containing 25 living genera with around 140 species distributed widely in the Indo-Pacific and Caribbean Sea (*Veron, 2000*; *Huang et al., 2014a*; *Huang et al., 2014b*). Since the landmark studies of *Fukami et al. (2004b)* and *Fukami et al. (2008)* that showed severe polyphyly involving four traditional families of Merulinidae Verrill (1865), Faviidae Gregory (1900), Pectiniidae Vaughan & Wells (1943) and Trachyphylliidae Verrill (1901), there has been considerable progress in building a phylogenetic classification of the clade (*Huang et al., 2009*; *Huang et al., 2011*; *Huang et al., 2014b*; *Budd & Stolarski, 2011*; *Benzoni et al., 2011*; *Arrigoni et al., 2012*; *Budd et al., 2012*). In particular, one subclade ('A' *sensu Budd & Stolarski, 2011*) comprising the genera *Merulina* Ehrenberg (1834), *Goniastrea* Milne Edwards & Haime (1848), *Scapophyllia* Milne Edwards & Haime (1848), and *Paraclavarina* Veron (1985), was analysed in detail using both nuclear and mitochondrial DNA markers to produce a well-supported phylogeny (Fig. 1). The groups
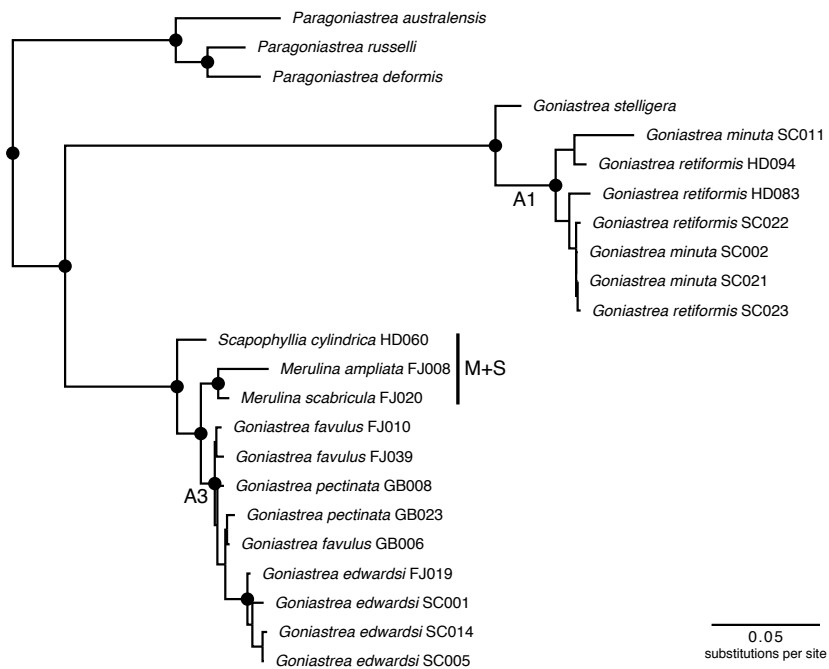

**Figure 1** **Molecular phylogenetic tree of Merulinidae corals.** Phylogeny based on *Huang et al. (2014a)*. Nodes at least moderately supported, with maximum likelihood and parsimony bootstrap >60 and Bayesian posterior probability >0.95, are labelled with circles. Host clades A1 and A3 follow the previous study, and are tested for differences in Symbiodiniaceae communities with *Merulina + Scapophyllia* (clade M + S).

defined by these supports (e.g., A1 and A3 *sensu* (*Huang et al., 2014a*) could be tested for phylogenetic conservatism of Symbiodiniaceae communities hosted by these corals.

Here, we apply DNA metabarcoding using the internal transcribed spacer 2 (ITS2) marker to test if closer lineages of merulinid corals host more similar communities and diversity of Symbiodiniaceae types (*Quigley et al., 2014*). We also test if the endosymbiont communities are structured according to conventional host taxonomy or geographic locality. The data show that endosymbiont community variation within host species overwhelms interspecific, intergeneric and among-lineage differences, and geography may play a role in structuring Symbiodiniaceae communities among closely-related corals.

## MATERIALS & METHODS
### Field sampling
Samples from eight merulinid species—*Goniastrea retiformis* (Lamarck, 1816), *G. pectinata* (Ehrenberg, 1834), *G. favulus* (Dana, 1846), *G. edwardsi* Chevalier, 1971, *G. minuta* *Veron, 2000*, *Merulina ampliata* (Ellis & Solander, 1786), *M. scabricula* Dana, 1846, and *Scapophyllia cylindrica* Milne Edwards & Haime, 1849—were collected from Singapore, Fiji, Seychelles and the Great Barrier Reef in Australia (Table S1). Field collections were approved by the National Parks Board Singapore (NP/RP16-156), Fiji Ministry of Education (21/10/10), Seychelles Bureau of Standards (A0347), and Great Barrier Reef Marine Park

Authority (G09/29715.1). In total, 30 healthy colonies were sampled across localities, with sampling at each locality performed within a maximum of one week, except for Singapore, for which samples were obtained in two batches in 2007 and 2017. Sampling periods were tracked to check if Symbiodiniaceae communities differed between them. Coral colonies were photographed in the field and a 10 to 100 cm$^2$ piece of tissue was sampled and preserved in 100% ethanol or CHAOS solution (4 M guanidine thiocyanate, 0.1% N-lauroyl sarcosine sodium, 10 mM Tris pH 8, 0.1 M 2-mercaptoethanol) (*Sargent, Jamrich & Dawid, 1986*; *Fukami et al., 2004a*; *Huang et al., 2008*).

## DNA extraction, amplification and sequencing

Isolation of digested material from CHAOS-preserved samples was performed with a phenol extraction buffer (100 mM TrisCl pH 8, 10 mM EDTA, 0.1% SDS), while ethanol-preserved samples were digested overnight with 900 μl CTAB buffer (cetyltrimethylammonium bromide; 0.1M Tris pH 8, 1.4M NaCl, 0.02M EDTA, 20 g/l CTAB) and 20 μl proteinase K (20 mg/ml). Phase separation was carried out using phenol:chloroform:isoamyl-alcohol (25:24:1), and precipitated DNA was eluted in 100–200 μl of water and stored at −80 °C.

Polymerase chain reaction (PCR) amplified a ∼350-bp fragment from the nuclear ribosomal internal transcribed spacer 2 (ITS2) region, targeted using a Symbiodiniaceae-specific primer set: ITSintfor2 (forward: 5′–GAA TTG CAG AAC TCC GTG–3′; *LaJeunesse & Trench, 2000*) and ITS-Reverse (reverse: 5′–GGG ATC CAT ATG CTT AAG TTC AGC GGG T–3′; *Coleman, Suarez & Goff, 1994*). This marker is located between the 5.8S and 28S ribosomal RNA genes, and has been informative for studying the diversity of Symbiodiniaceae (e.g., *LaJeunesse, 2001*; *LaJeunesse, 2002*; *LaJeunesse & Thornhill, 2011*; *Arif et al., 2014*; *Boulotte et al., 2016*). A unique 8-bp barcode was added to the 5′end of each primer for multiplexed amplicon sequencing. In total, 100 sets of tagged primers were used for 90 PCR reactions (triplicates of 30 samples) and 10 no-template controls. Each reaction mixture comprised a total volume of 25.0 μl, including 12.5 μl GoTaq DNA polymerase (Promega Corporation, Madison, WI, USA), 1.0 μl 10 μM primers (forward and reverse), 1.0 μl DNA template and 9.5 μl water. The PCR profile consisted of an initial denaturation of 94 °C for 180 s, following by 35 cycles of 94 °C for 30 s, 55 °C for 30 s and 72 °C for 45 s, and a final extension step of 72 °C for 180 s.

The quality of PCR products was assessed with electrophoresis on a 1% agarose gel stained with GelRed (Cambridge Bioscience, Cambridge, UK) and visualised under ultraviolet exposure. Gel band intensities were used to approximate normalisation during pooling of amplicons. The pooled library was purified using 1.0 × Sera-Mag SpeedBeads (GE Healthcare Life Sciences) in 18% polyethylene glycol (PEG) buffer suspension:DNA ratio and finally eluted with 30 μl of water. The amount of DNA post-purification was quantified using a Qubit Fluorometer 3.0 (Life Technologies) and made up to 20 ng/μl, with 50 μl sent to Axil Scientific Pte Ltd for DNA library preparation using the TruSeq DNA PCR-Free Library Prep Kit (Illumina). Sequencing was carried out in ∼50% of an Illumina MiSeq run, using the Reagent Kit v3 for 300-bp paired-end reads. All sequences generated here are available at the National Center for Biotechnology Information (NCBI; BioProject ID: PRJNA549817).

## Data analysis

The bioinformatics pipeline was based primarily on *Sze et al. (2018)*, *Afiq-Rosli et al. (2019)* and *Lim et al. (2019)* (but see *Hume et al., 2019*). Briefly, Paired-End reAd mergeR (PEAR) (version 0.9.8) (*Zhang et al., 2014*) was used to assemble paired-end sequences with a default minimum overlap of 100 bp; minimum and maximum lengths of 300 and 390 bp respectively, based on the expected amplicon size; and a minimum Phred quality score of 30 to ensure accurate base calls. OBITools (version 1.2.11) (*Boyer et al., 2016*) were used to process the assembled paired-end reads. Reads were demultiplexed using *ngsfilter* based on the primer sequences, unique tags and sample information to obtain sample sequences. *obiuniq* collapsed identical sequences and assigned read counts for each unique DNA sequence. *obisplit* sorted sample data into separate files based on their sample names. Sequences with only a single read were discarded using *obigrep*. Finally, *obiclean* was used to identify amplification and sequencing errors by assigning sequence records as 'head' (most common sequence amongst all the sequences), 'internal' (erroneous sequences) and 'singleton' (rare, likely true sequences with no other variants). Sequences assigned as 'head' were retained for further analysis. No sequences were found in the negative PCR controls following quality checks and filtering.

Sequences were searched against the NCBI *nt* database using BLAST+ (version 2.3.0; *blastn*) (*Altschul et al., 1990*), with maximum *E*-value of $10^{-6}$ and minimum identity match of 85%. Identity of each sequence was established based on the top match of Symbiodiniaceae type with ≥97% identity, in accordance with previous studies using the ITS2 marker (*Arif et al., 2014*; *Cunning, Gates & Edmunds, 2017*). Following sequence filtering and taxon identification, two endosymbiont community datasets were assembled for subsequent analyses: (1) types with minimum of 10 reads summed across PCR triplicates for each sample; and (2) types present in at least two of the three PCR replicates for each sample. These strict filtering steps would help reduce false positives associated with amplification and sequencing errors. The datasets are available at Zenodo (http://dx.doi.org/10.5281/zenodo.3344613).

The host phylogeny was reconstructed in *Huang et al. (2014a)* and pruned to show only the coral samples used here (Fig. 1). Briefly, phylogenetic analyses were carried out on the concatenated matrix comprising the nuclear histone H3 (*Colgan et al., 1998*), internal transcribed spacers 1 and 2 (*Takabayashi et al., 1998a*; *Takabayashi et al., 1998b*), and mitochondrial non-coding intergenic region (*Fukami et al., 2004a*). The maximum likelihood tree was inferred in RAxML 7.7.9 (*Stamatakis, 2006*; *Stamatakis, Hoover & Rougemont, 2008*) with 50 random starting trees and 1,000 bootstrap replicates. Bayesian inference was carried out in MrBayes 3.2.2 (*Huelsenbeck & Ronquist, 2001*; *Ronquist & Huelsenbeck, 2003*; *Ronquist et al., 2012*) with four Markov chains of 6 million generations implemented in two runs, logging one tree per 100 generations and discarding the first 10,001 trees as burn-in. Maximum parsimony analysis was performed in PAUP* 4.0b10 (*Swofford, 2003*) with 10,000 random additions and 1,000 bootstrap replicates.

Endosymbiont data analysis was performed in R (version 3.4) (*R Core Team, 2013*) using the *vegan* package (version 2.4.6) (*Oksanen et al., 2013*). Variation of Symbiodiniaceae communities among the coral colonies was examined using non-metric multidimensional

scaling (NMDS) with the Bray-Curtis dissimilarity—based on proportional read abundance—and Jaccard distance measures. Analysis of similarities (ANOSIM; 999 permutations) was performed to test for differences in the proportional read abundances of Symbiodiniaceae types among the host coral lineages (A1, A3 and M+S; Fig. 1), host genera (*Goniastrea*, *Merulina* and *Scapophyllia*), and host localities (Australia, Fiji, Singapore and Seychelles), independently. The taxonomic distinctness index was also computed to characterise patterns of Symbiodiniaceae assemblage dissimilarity among host species (*Clarke & Warwick, 1998*; *Warwick & Clarke, 2001*).

## RESULTS & DISCUSSION

A total of 10,496,267 reads have been obtained from the MiSeq sequencing run, with 10,119,346 paired-end reads assembled (96.4%). These reads correspond to 646,568 unique haplotypes. Following error pruning and sequence filtering, 294 haplotypes remain, belonging to three major Symbiodiniaceae genera that are common in Indo-Pacific corals—*Symbiodinium* (formerly clade A), *Cladocopium* (formerly clade C) and *Durusdinium* (formerly clade D) (*LaJeunesse et al., 2010a*; *LaJeunesse et al., 2018*; *Yang et al., 2012*).

*Cladocopium* is overall the most abundant endosymbiont, accounting for 78.2% of sequences retained, and furthermore the most dominant taxon based on proportional read abundance in six of the eight host species examined (Fig. 2). *Cladocopium* is present in all host species, while *Durusdinium* and *Symbiodinium* are present in six and three host species, respectively. Our results mirror previous community-level studies of Symbiodiniaceae showing that *Cladocopium* is more abundant in tropical latitudes compared to other clades such as *Symbiodinium* which is mostly found at higher latitudes (*Rodriguez-Lanetty et al., 2001*; *Savage et al., 2002*; *Baker, 2003*). In particular, endosymbiont communities in corals from the tropical Pacific and Southeast Asian regions appear to be dominated by *Cladocopium* and, to a lesser extent, *Durusdinium* (*Rodriguez-Lanetty et al., 2001*; *Van Oppen et al., 2001*; *Baker, 2003*; *LaJeunesse et al., 2010a*; *Tanzil et al., 2016*). While there are distinct patterns in read abundances of Symbiodiniaceae genera among coral species, proportional read count variance between samples of each host species is large (Fig. 2), revealing large intraspecific variation of endosymbiont abundances. For example, among *G. pectinata* colonies, *Durusdinium* reads are on average more abundant than *Symbiodinium* reads, but *Durusdinium* is not present in three colonies examined. Therefore, at the Symbiodiniaceae genus level, variation among coral colonies within each species may mask differences between species.

Twenty-three Symbiodiniaceae ITS2 types have been found across all colonies, comprising three types from *Symbiodinium*, eighteen from *Cladocopium* and two from *Durusdinium*. *Cladocopium goreaui* (ITS2 type C1) and *Cladocopium* C3 are the most common types, present in 32% and 55% of all colonies examined. The latter is also present in every merulinid species except *Goniastrea minuta*, corroborating past studies showing that *Cladocopium* C3 is a generalist type in the Indo-Pacific (*LaJeunesse et al., 2003*; *Wham, Carmichael & LaJeunesse, 2014*), prevalent even in some of the world's warmest reefs
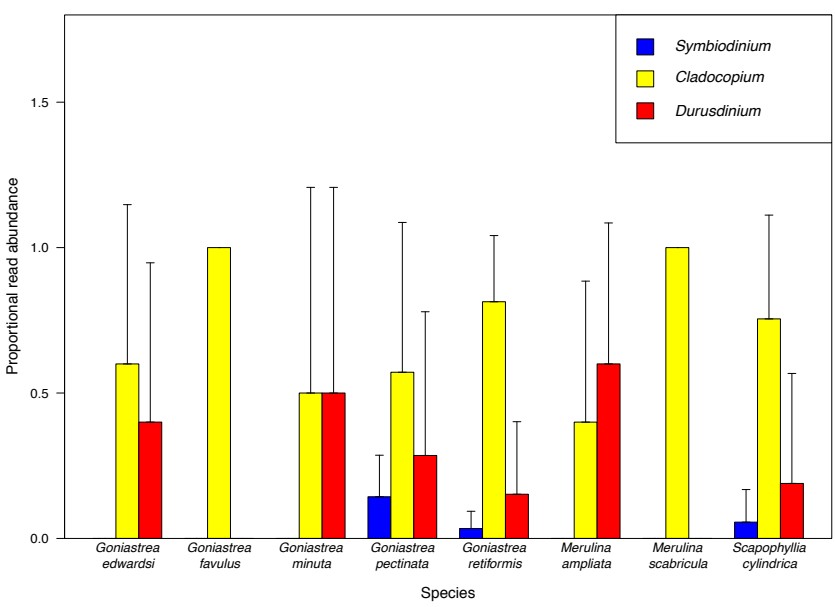

**Figure 2 Proportional abundance of sequencing reads recovered.** MiSeq sequencing data for each Symbiodiniaceae genus among coral species examined.

(*Hume et al., 2013*). Similarly, *Cladocopium goreaui* (ITS2 type C1) is also considered to be a generalist taxon worldwide (*LaJeunesse et al., 2003*; *LaJeunesse, 2005*; *Wham, Carmichael & LaJeunesse, 2014*).

Among the coral species examined here, *Goniastrea pectinata* and *Scapophyllia cylindrica* contain the richest Symbiodiniaceae assemblages of 11 and 10 types respectively, with *Goniastrea retiformis*, *Goniastrea favulus* and *Merulina ampliata* following with nine different types. *Merulina scabricula* only hosts two Symbiodiniaceae types despite being a congener and clustering most closely with *M. ampliata* (Fig. 1), which has nine types.

Endosymbiont community structure does not differ among host genera ($R = -0.180$, $p = 0.979$) and lineages ($R = 0.085$, $p = 0.142$; as defined in Fig. 1) when analysing sequences based on minimum of 10 reads summed across PCR triplicates for each sample (Fig. 3). Results are the same for types present in at least two of the three PCR replicates for each sample ($R = -0.110$, $p = 0.792$ for genera; $R = 0.110$, $p = 0.146$ for lineages; Fig. S1). Furthermore, no genus- or lineage-specific patterns of endosymbiont-type taxonomic distinctness were apparent among host taxa (Fig. S2). In support, although *Goniastrea retiformis* and *G. minuta* are in a lineage distinct from the rest of the species examined (Fig. 1), their endosymbiont communities are neither clustered nor separated from the other species on the NMDS (Fig. 3). Symbiodiniaceae communities does appear to be structured spatially as they are significantly different between Australia, Fiji, Singapore and Seychelles ($R = 0.162$, $p = 0.036$), though this pattern is not significant when considering the types present in at least two of the three PCR replicates ($R = 0.054$, $p = 0.313$). These results should therefore be interpreted with caution.
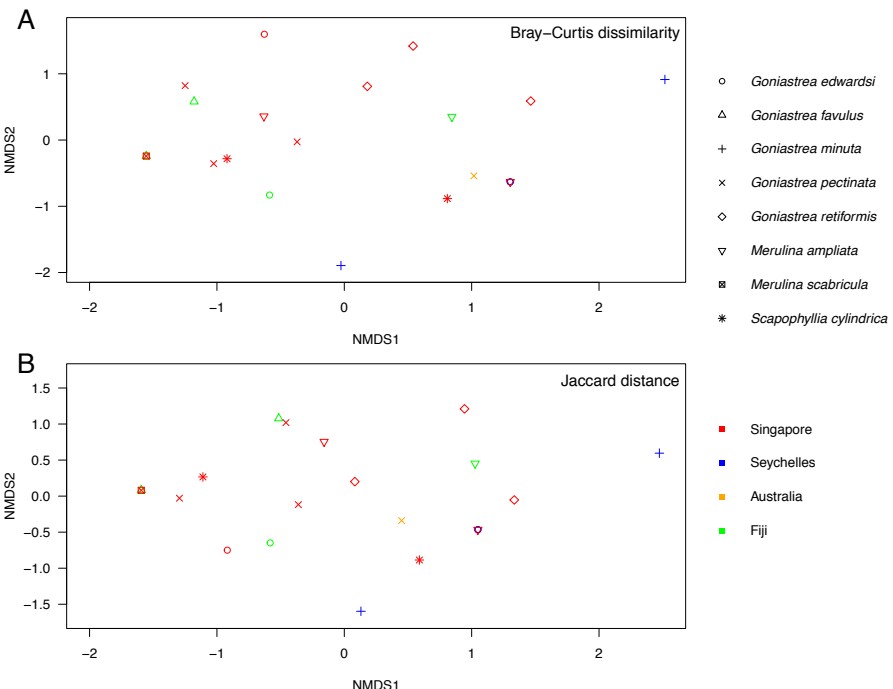

**Figure 3 Non-metric multidimensional scaling of Symbiodiniaceae communities.** Analysis of endosymbiont communities in Merulinidae corals based on the Bray–Curtis dissimilarity (A; stress = 0.097) and Jaccard distance (B; stress = 0.093) measures of symbiont types with ≥10 reads in each colony. Symbols are distinguished according to host species and locality.

Host species identity has been shown to play a role in determining endosymbiont community composition (*Thornhill et al., 2009*; *Thornhill et al., 2014*; *Tonk et al., 2013*). For example, *Siderastrea siderea* in the Belize Mesoamerican Barrier Reef System contains a distinct Symbiodiniaceae community signal (*Baumann et al., 2018*). This host specificity reflects various interacting physiological needs of the coral species, such as their depth, light, temperature or nutrient preferences (*Finney et al., 2010*), and the endosymbiont populations can be further structured according to the environment (*Thornhill et al., 2009*; *Davies et al., 2019*). Interestingly, the host species signal breaks down at the genus level, as Symbiodiniaceae communities in *Siderastrea radians* are more similar to *Pseudodiploria strigosa* than its congeneric *Siderastrea siderea* (*Baumann et al., 2018*).

Indeed, various combinations and diversity of endosymbionts allow the holobiont to face distinct environmental stressors (*Toller, Rowan & Knowlton, 2001a*; *Toller, Rowan & Knowlton, 2001b*; *Berkelmans & Van Oppen, 2006*; *LaJeunesse et al., 2009*). For instance, while the presence of endosymbiotic *Cladocopium* is associated with large depth ranges and stable growth rates of coral hosts (*Little, Van Oppen & Willis, 2004*; *Cantin et al., 2009*), this genus and in particular the supposed generalist *Cladocopium* C3 can be sensitive to thermal variation at local scales (*Keshavmurthy et al., 2012*). Therefore, the intraspecific variation in community structure shown in each of the species examined here may be related to environmental differences among host localities (*Howells et al., 2012*). Because

our samples originate from a large spatial range, and since *Cladocopium* can be found throughout the Indo-Pacific realm (*LaJeunesse et al., 2003*; *Howells et al., 2012*; *Wham, Carmichael & LaJeunesse, 2014*; *Wong et al., 2016*; *Keshavmurthy et al., 2017*), our sampling is expected to capture a considerable fraction of the environmental affinities associated with this Symbiodiniaceae genus.

*Durusdinium* is present in six host species despite being represented by lower read abundance compared with *Cladocopium*. This is not surprising given that it is the second most-dominant clade behind *Cladocopium* in the tropical Indo-Pacific (*Rodriguez-Lanetty et al., 2001*; *Van Oppen et al., 2001*; *Baker, 2003*; *LaJeunesse et al., 2010a*; *Tanzil et al., 2016*). Importantly, *Durusdinium* is considered an opportunistic taxon, increasing in abundance *in hospite* during challenging conditions such as coral bleaching, and more generally under extreme or variable levels of heat or light (*Toller, Rowan & Knowlton, 2001a*; *Chen et al., 2003*; *Lien et al., 2007*; *Lien et al., 2013*). Therefore, the presence and abundance, or absence, of *Durusdinium* could serve as an indicator of coral community health (*Warner, Fitt & Schmidt, 1999*; *Burnett, 2002*; *Baker et al., 2004*; *Reynolds et al., 2008*).

The high light tolerance of *Durusdinium* in shallow water environments may underlie the capacity of colonies living on the reef flat that are often exposed to elevated irradiance levels (see *Innis et al., 2018*). These corals consist of species such as *Goniastrea retiformis*, *G. minuta* and *G. pectinata* (*Huang et al., 2006*; *Ng, Chen & Chou, 2012*) which we have found to host *Durusdinium* (Fig. 2). Furthermore, this endosymbiont genus tolerates the warmer periods during the May-June inter-monsoon when the water temperature is in excess of 30 °C (*Sin et al., 2016*; see *Berkelmans, 2002*). In more seasonal localities, the summer may bring about the increased dominance of *Durusdinium* in symbiosis among certain taxa such as *Acropora* (*Chen et al., 2005*; *Berkelmans & Van Oppen, 2006*; *Yorifuji et al., 2017*), although the degree of symbiont shuffling is affected by each colony's thermal history (*Hsu et al., 2012*). Here, despite being collected during different time periods throughout the year and over time (Table S1), there are no patterns to suggest discernible seasonality in endosymbiont composition in the merulinids, but further temporal sampling at each study site is needed to uncover any seasonal variations.

## CONCLUSIONS

Overall, our results show that Symbiodiniaceae communities are not clustered according to coral host species or genera, and show no discernible patterns based on the host phylogeny. Clearly, intraspecific differences dominate the variation among corals in both endosymbiont community structure and diversity, with no distinct communities hosted by particular coral species or genera. We posit that the complexity related to horizontal endosymbiont acquisition among the broadcast-spawning coral species examined here could contribute to this (*Baird, Guest & Willis, 2009*; *Hartmann et al., 2017*). Even as heritability of symbionts has been shown to be substantial in some species (*Quigley, Willis & Bay, 2017*), it is not uncommon for a coral host to harbour distinct Symbiodiniaceae communities over its life cycle. For example, juvenile hosts of *Acropora* corals contain a more diverse Symbiodiniaceae community compared to their adult stage

(*Cumbo, Baird & Van Oppen, 2013*; *Suzuki et al., 2013*; *Yamashita et al., 2013*). Subsequent dispersal to areas with differing backgrounds of Symbiodiniaceae diversity (*Nitschke, Davy & Ward, 2016*) may further enhance this complexity as uptake of environmental Symbiodiniaceae or community changes induced under the new conditions could drive shifts in endosymbiotic composition (*Quigley, Willis & Bay, 2017*). To a limited extent, coral colonies examined here do show consistent patterning of Symbiodiniaceae communities based on their areas of origin. More extensive collections of corals from various regions and environments will help us better understand the specificity of the coral-endosymbiont relationship.

Large-scale spatial structuring of endosymbionts has been hypothesised during the earliest stages of such research (*Baker & Rowan, 1997*; *Rowan, 1998*) and also demonstrated in studies across a diverse range of invertebrates (*LaJeunesse et al., 2004*; *Gong et al., 2018*). Today, even with the application of new sequencing technologies that confer higher sensitivity for detecting Symbiodiniaceae types, the resolution of spatial patterns remains rough due to sparse sampling. Certainly, a concerted sampling effort at more reef localities will help verify our findings and provide new insights into the distribution and biogeography of this vital symbiosis.

## ACKNOWLEDGEMENTS

Computational work was partially performed with resources from the National Supercomputing Centre, Singapore (https://www.nscc.sg).

### Funding

This work was supported by the National Research Foundation, Prime Minister's Office, Singapore under its Marine Science R&D Programme (MSRDP-P03). The funders had no role in study design, data collection and analysis, decision to publish, or preparation of the manuscript.

### Grant Disclosures

The following grant information was disclosed by the authors:
National Research Foundation, Prime Minister's Office: MSRDP-P03.

### Competing Interests

The authors declare there are no competing interests.

### Author Contributions

- Sébastien Leveque performed the experiments, analyzed the data, prepared figures and/or tables, authored or reviewed drafts of the paper, approved the final draft.
- Lutfi Afiq-Rosli, Yin Cheong Aden Ip and Sudhanshi S. Jain contributed reagents/materials/analysis tools, authored or reviewed drafts of the paper, approved the final draft.

- Danwei Huang conceived and designed the experiments, contributed reagents/materials/analysis tools, prepared figures and/or tables, authored or reviewed drafts of the paper, approved the final draft.

## Field Study Permissions

The following information was supplied relating to field study approvals (i.e., approving body and any reference numbers):

Field collections were approved by the National Parks Board Singapore (NP/RP16-156), Fiji Ministry of Education (21/10/10), Seychelles Bureau of Standards (A0347), and Great Barrier Reef Marine Park Authority (G09/29715.1).

## DNA Deposition

The following information was supplied regarding the deposition of DNA sequences:

All sequences are available at NCBI: BioProject ID PRJNA549817.

## Data Availability

The datasets are available at Zenodo: Leveque, Sébastien, Afiq-Rosli, Lutfi, Ip, Yin Cheong Aden, Jain, Sudhanshi S., & Huang, Danwei. (2019). Searching for phylogenetic patterns of Symbiodiniaceae community structure among Indo-Pacific Merulinidae corals [Data set]. PeerJ. Zenodo. http://doi.org/10.5281/zenodo.3344613

## Supplemental Information

Supplemental information for this article can be found online at http://dx.doi.org/10.7717/peerj.7669#supplemental-information.

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
