# Peer review of "Searching for phylogenetic patterns of Symbiodiniaceae community structure among Indo-Pacific Merulinidae corals"

_PeerJ, doi:10.7717/peerj.7669_

## Round 0.1 · original submission · Major Revisions

Three expert reviewers have evaluated your manuscript and their comments can be seen below and in a PDF. The research presented here is original, the study question is well defined and the data set is potentially very interesting. The manuscript is well written and structured. As pointed out in the PDF there are some additional relevant papers that should be incorporated.

However, the methods require additional details to fully understand the analysis of sequencing data. Also, data analysis needs to be redone (e.g. analysis and presentation of proportional abundance instead of sequencing count data to avoid bias) and important data and results need to be included in the paper (e.g. include type level data; share summary and/or raw sequencing data). These issues can be addressed without any further field or lab work.

Reviewer 1 ·

Basic reporting

The paper is generally well written, although there are some additional relevant papers that should be incorporated in the Discussion. I have noted these in the annotated pdf review. The figures present sequencing count data, however this is biased by the different numbers of sequences obtained for each sample, and should be presented (and analysed) as proportional abundance. Additionally symbiont data are presented at the genus level only and the type level data are missing. Furthermore, summary and/or raw sequencing data have not been shared. Consequently, I cannot evaluate the validity of the study findings.

Experimental design

The research is original and the study question is well defined. The methods are mostly sufficient but additional details are required for the reader to fully understand the analysis of sequencing data. Please see comments in the annotated pdf review.

Validity of the findings

Based on limitations with the current data analysis and presentation I cannot evaluate the validity of the findings.

Additional comments

Potentially you have a very interesting data set. However this is not obvious to the reader as key elements are not presented in the paper. For this study to be publishable, the sequencing data need to be reanalysed and the symbiont type data need to be presented. I also strongly encourage you to see how your results look using the symportal analytical framework (http://symportal.org/).

Annotated reviews are not available for download in order to protect the identity of reviewers who chose to remain anonymous.

Reviewer 2 ·

Basic reporting

The paper surveyed endosymbiont diversity and community structure in merulinid corals and report large intraspecific variation, as well as the apparent influence of geography on community structure. The paper is written in clear language, and is well-organised. The introduction and background is sufficient and concise, providing context to the paper’s objectives. The research question is well-defined. Figures and tables are well-presented.

Experimental design

The research question is well defined, relevant and meaningful in the context of biological diversity studies and to some extent global change biology. Ethical standards for field collection and sampling, and technical standards for methodology and data analysis conform to high standards. The methods (including data analysis) are described with sufficient detail and are appropriate.

Validity of the findings

The data is robust and self-contained. Overall, the conclusions are well-stated, with cautionary statements on certain aspects where the data provide limited support, i.e. lines 254-258.The spatial scale of the paper is commendable, but therein also lies its limitations which the authors rightly acknowledge, i.e. the lack of finer scale sampling which may provide better insight into patterns driving community structure at smaller spatial scales.

Reviewer 3 ·

Basic reporting

Searching for phylogenetic patterns of Symbiodiniaceae community structure among Indo Pacific corals written by Leveque et al examined Symniodiniaceae communities of 8 Merulinid species collected from 4 different locations in the Indo-Pacific by sequencing ITS2 using NGS. Based on the non-metric multidimensional scaling with the Bray-Curtis dissimilarity and Jaccard distance measures, authors found Symbiodiniaceae communities are not clustered according to coral species or genera and/or their host phylogenetic patterns, though intra-species patterns dominate the variation. The wide-range of Indo-Pacific sampling of phylogenetically well-defined coral spices enable the hypothesis testing proposed in the introduction. The result presented here is contrasting with the previous study of Pocillopora and worth publishing.
Overall the manuscript is well-structured and well-written. I just would like to suggest several points to meet PeerJ’s criteria.

1. Raw data for MiSeq run should be deposited and opened before acceptance.
2. There is no materials and methods about Fig. 1 (even if the tree is from Huang et al 2014b, it will be better to open the FASTA data and briefly describe material and methods). It will be helpful to have material and method information on them at least in the supplementary material.
3. Regarding the spatial structure of Symbiondiniaceae, citing contrasting results by Thornhill D, (Xiang Y, Fitt W, Santos S (2009) Reef endemism, host specificity and temporal stability in populations of symbiotic dinoflagellates from two ecologically dominant Caribbean corals. PLoS ONE, 4, e6262.) would be also useful.

The title is very attractive but it will be better include “Merulinid or Merulinidae” because not all Indo Pacific coral species have been examined.

Experimental design

Overall there is no significant problem
But authors should be careful about using Go-taq because the sequences obtained using this taq contain a lot of artificial errors (1 bp over 300 bp) and caution is required especially when using NGS.
Please briefly describe how these errors can be ignored in this study (I think this does not affect the large classification used in this study. But not sure).
L188-L200 Please briefly describe the rationale for using these criteria (97% identity etc)
L204-209 The analysis conducted in this study is good enough to deduce the conclusions, but it might be also useful to use “taxonomic distinctness index”. Warwick RM, Clarke KR 2001. Practical measures of marine biodiversity based on relatedness of species. Ocean.Mar.Biol.Ann.Rev. 39, 207-231

Validity of the findings

L304 Overall, our results show that coral communities
→coral Symbiodiniaceae communities?
Fig.2 How do Arthurs interpret the relatively huge error bars in Cladocopium for G. favulus and G. retiformis)?

Additional comments

nothing

---

## Round 0.2 · accepted · Accept

I am satisfied with the changes that have been made to the resubmitted manuscript.

Reviewer 3 ·

Basic reporting

The paper has been greatly improved especially following the recommendation by the reviewer1's comment to use "proportional abundance".
I'm happy with the revised version and I think it worth publishing.

Experimental design

Improved and met the standards

Validity of the findings

No comment

Additional comments

Please take care of ABS when you use genetic materials from abroad in the future.